# TVTSyn: Content-Synchronous Time-Varying Timbre for Streaming Voice Conversion and Anonymization

**Waris Quamer, Mu-Ruei Tseng, Ghady Nasrallah & Ricardo Gutierrez-Osuna**
Department of Computer Science and Engineering
Texas A&M University
College Station, TX 77840, USA
{quamer.waris,mtseng,ghadynasrallah,rgutier}@tamu.edu

## Abstract

Real-time voice conversion and speaker anonymization require causal, low-latency synthesis without sacrificing intelligibility or naturalness. Current systems have a core representational mismatch: content is time-varying, while speaker identity is injected as a static global embedding. We introduce a streamable speech synthesizer that aligns the temporal granularity of identity and content via a content-synchronous, time-varying timbre (TVT) representation. A Global Timbre Memory expands a global timbre instance into multiple compact facets; frame-level content attends to this memory, a gate regulates variation, and spherical interpolation preserves identity geometry while enabling smooth local changes. In addition, a factorized vector-quantized bottleneck regularizes content to reduce residual speaker leakage. The resulting system is streamable end-to-end, with <80 ms GPU latency. Experiments show improvements in naturalness, speaker transfer, and anonymization compared to SOTA streaming baselines, establishing TVT as a scalable approach for privacy-preserving and expressive speech synthesis under strict latency budgets.

## 1 Introduction

Real-time voice conversion (VC) and speaker anonymization (SA) aim to deliver natural, intelligible speech while meeting strict streaming and latency constraints. Beyond words, voice recordings carry biometric and paralinguistic cues–identity, sex, age, accent, and emotion–that adversaries can exploit for recognition and profiling, creating real risks to privacy. As voice interfaces proliferate and privacy rules tighten, protecting these speech attributes without degrading the communicative utility of speech has become essential, as exemplified by several initiatives. As an example, starting in 2020 the Voice Privacy Challenge (Tomashenko et al., 2020) evaluates speech anonymization systems on both privacy and usefulness, with shared benchmarks that quantify speaker obfuscation alongside speech quality. Further, federal agencies have pushed for the development of real-time solutions with tight latency budgets (e.g., IARPA's Anonymous Real-Time Speech program [1]).

To address this challenge, recent speech architectures have achieved sub-second latency by combining lightweight content encoders with direct waveform decoders (Quamer & Gutierrez-Osuna, 2024; 2025a; Yang et al., 2022; Chen et al., 2023). Yet a core limitation of these approaches persists: while content is represented as a time-varying sequence, speaker identity is typically injected as a single static vector. This dynamic-static mismatch dampens expressivity and often yields over-smoothed timbre, especially when articulation, emotion, or emphasis change within an utterance. Making content strongly speaker-independent (e.g., via aggressive bottlenecks) can improve anonymization performance but suppress meaningful variations in speech such as accent and emotional color, or introduce artifacts (Quamer & Gutierrez-Osuna, 2025a). We contend this trade-off is *largely* architectural: a stationary speaker vector forces the decoder to reconcile incompatible time scales. A better formulation would add temporal granularity to speaker conditioning to match that of the content, while allowing control and meeting tight latency constraints.

---

[1] https://www.iarpa.gov/research-programs/arts

We propose TVTSyn, a streaming speech synthesizer that replaces static speaker embeddings with a time-varying timbre (TVT) representation that is synchronized with the content. A Global Timbre Memory (GTM) expands a global timbre seed into a compact set of timbre facets; frame-level content attends to this memory to retrieve the most relevant facets over time; a learned gate regulates how much timbre is allowed to vary; and spherical interpolation blends global and time-varying paths to preserve identity geometry while enabling smooth local variation. This TVT stream conditions a causal decoder alongside pitch/energy predictors, yielding natural variation while retaining control. Complementing this, a factorized vector-quantized bottleneck regularizes the content network to reduce residual identity cues while preserving linguistic content. TVTSyn runs in a streaming fashion with small, mask-based future access in the encoder and fully causal decoding. The system can generate synthesis with $< 80ms$ latency on a modern GPU and runs within a few hundred ms. latency on CPUs[2]. We evaluate the model for both VC and SA tasks under the VoicePrivacy Challenge (VPC) 2024 protocol, reporting equal-error-rates (EER) in automatic speaker verification (ASV) as a measure of privacy preservation, and word error rates (WER) of an automatic speech recognizer (ASR) as a measure of utility, as well as latency and real-time factors. Our main contributions are:

- **Content-synchronous timbre modeling**: We introduce a time-varying timbre formulation that aligns speaker conditioning with frame-level content, resolving the static–dynamic mismatch responsible for degraded quality in streaming VC and SA.

- **A streamable low-latency architecture**: We design a fully causal system that integrates GTM-based timbre, factorized VQ bottlenecks, and prosodic predictors, and maintains low latency while balancing naturalness with speaker fidelity, and anonymization robustness.

- **Comprehensive benchmarking**: We evaluate across VC and anonymization tasks with perceptual quality, speaker similarity, privacy (EER), utility (WER), and runtime performance, demonstrating superior privacy–utility trade-offs over prior streaming systems.[3]

## 2 RELATED WORK

**Voice conversion (VC).** Conventional VC models decompose the speech signal into content and speaker channels, then re-synthesize the content channel with a target voice identity. Early pipelines relied on cascaded ASR-TTS (text-to-speech) modules or used phonetic posteriorgram (PPG) representations to obtain a speaker-independent content stream before conditioning a multi-speaker decoder on the target voice (Huang et al., 2020; Liu et al., 2021). To avoid the brittleness of text/PPG representations, later "disentanglement" models learned content directly from audio via information bottlenecks and normalization/MI penalties, or with discrete units (e.g., VQ, HuBERT pseudo-labels) that suppress residual speaker cues while keeping phonetic detail (Chan et al., 2022; Chen et al., 2021; Wang et al., 2021; Quamer et al., 2023; Quamer & Gutierrez-Osuna, 2025b). Self-supervised speech representations such as HuBERT have since become standard for providing robust, transcript-free supervision of content features (van Niekerk et al., 2022).

Streaming VC introduces strict latency constrains. Supervised systems such as LLVC (Sadov et al., 2023) can achieve latencies as low as 20 ms, but their reliance on parallel data makes them difficult to scale. By contrast, most unsupervised VC approaches adopt causal encoders with small future peeks, sliding-window inference, and fast waveform decoders, achieving sub-second latency while maintaining intelligibility (Quamer & Gutierrez-Osuna, 2024; 2025a; Yang et al., 2022; Chen et al., 2023; Zhang et al., 2025). Here, speaker identity is usually a single global embedding injected at every frame–by concatenating content and speaker embeddings, and more recently via FiLM/AdaIN or conditional layer normalization (CLN), which allow each channel to be scale/shift modulated, enabling few/zero-shot adaptation in multi-speaker TTS/VC (Huang & Belongie, 2017; Perez et al., 2018). A complementary line models intra-speaker variability with multi-center (sub-center) training, learning multiple anchors per speaker to reduce intra-class scatter and capture channel/phonetic/affective variation (Ulgen et al., 2024). However, these streaming approaches typically use static speaker embeddings that remain constant across all frames. While this enables low latency, it creates a representational mismatch with dynamic content embeddings. Recent attention-based methods attempt to address this: FreeVC (Li et al., 2023) uses static embeddings lacking temporal variation; GenVC (Cai et al., 2025) employs learned queries but requires non-causal inference; DAFMSVC (Chen et al., 2025) enables content-aware modulation but is offline-only and lacks

---

learnable speaker prototypes. Our GTM differs by introducing learnable prototype parameters that capture universal timbre characteristics across speakers, while speaker-specific modulation adapts them to individual identities—enabling efficient generalization under streaming constraints.

**Speaker anonymization (SA).** Anonymization aims to mask speaker identity while preserving communicative utility. Traditional digital-signal-processing (DSP) approaches manipulate the signal through formant shifting (e.g., McAdams coefficient (McAdams, 1984)), frequency warping, vocal-tract length normalization, pitch/rate modifications, or modulation spectrum smoothing (Patino et al., 2021; Tavi et al., 2022). Though these training-free approaches are lightweight and fast, they struggle against modern ASV back-ends. Starting in 2020, the VPC formalized attacker models and popularized two baselines: a DSP approach (McAdams), and a machine-learning (ML) pipeline with x-vectors plus neural synthesis–solidifying the quality-privacy trade-off as part of the evaluation protocols (Tomashenko et al., 2020).

ML-based anonymization typically follows the VC template: extract content, replace identity with an anonymized embedding, then synthesize. Identity replacement includes farthest-speaker selection (Srivastava et al., 2020), autoencoder-based removal of protected attributes Perero-Codosero et al. (2022), codebook/lookup strategies Hsu et al. (2018), and learned pseudo-speaker generators (GAN-based sampling in embedding space) Meyer et al. (2023). Recent systems underscore the need to keep distinctiveness and emotional color while pushing EERs towards chance levels. Streaming variants compress encoders and remove heavy SSL/ASR stacks to meet latency constraints, sometimes allowing limited future context in causal attention layers (Quamer & Gutierrez-Osuna, 2024; 2025a). Discrete bottlenecks further anonymize content but can hurt naturalness if too aggressive—highlighting the need to align the temporal granularity of identity conditioning with frame-synchronous content. Finally, language-model-based generative VC/anonymization such as GenVC demonstrates strong style transfer by tokenizing phonetic and acoustic streams and conditioning a causal LM with a style prompt before vocoding; these models show the benefits of better temporal modeling and rich style prompts, though most target offline applications (Cai et al., 2025).

## 3  METHODS

Shown in Figure 1b, our system architecture consists of four modules: (1) a content encoder that generates discrete, speaker-independent linguistic representations in a causal manner, (2) a speaker processing block that consumes global speaker embeddings and produces content-aligned, time-varying timbre representations, (3) pitch and energy predictors that model frame-level prosodic variation, and (4) a decoder that synthesizes speech waveforms directly from the combined representation. At inference, the framework can convert/anonymize speaker identity by altering the speaker embedding or its time-varying trajectory, while leaving linguistic information intact.

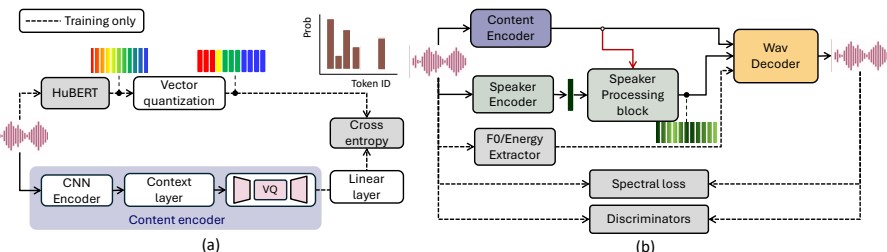

Figure 1: (a) The content encoder in TVTSyn is trained separately with supervision from an off-line HuBERT model. (b) The waveform decoder is trained in a self-supervised fashion to reconstruct the input utterance from content and speaker embedding streams. Dashed lines are disabled at inference.

### 3.1  STREAMING CONTENT ENCODER

**Feature extraction.** The content encoder transforms input waveforms into 512-dim. frame-level embeddings that emphasize linguistic content while suppressing speaker-specific cues. It is implemented as a fully causal 1-D CNN followed by a contextual self-attention layer. The CNN begins with an initial convolution (kernel size 7) to capture fine-scale waveform structure, and then applies four downsampling stages with stride ratios of [8, 5, 4, 2], yielding an overall hop of 320 samples (∼20 ms at 16 kHz). Each stage doubles the channel width and is preceded by a lightweight residual

block with two convolutions (kernels 3 and 1, dilation 2) and a true skip connection, preserving local phonetic detail while remaining streamable. A final convolution (kernel size 3) projects the sequence into a 512-dim. latent space. To capture dependencies beyond the CNN's local receptive field, we append a stack of 8 causal multi-head self-attention (MHSA) blocks with a fixed look-back window of $W=2$ s. To avoid using a separate look-ahead module (Quamer & Gutierrez-Osuna, 2025a), TVTSyn expands the causal attention mask to up to 4 future tokens ($\sim$80 ms). Formally, at time step $t$, attention is masked to keys/values from $\{t-\tau, \ldots, t, \ldots, t+4\}$ with $\tau$ corresponding to $W$. This provides stable long-range temporal coherence with short-term anticipatory cues for coarticulation while avoiding the latency overhead of a separate look-ahead module. During inference, a ring KV cache maintains a rolling $\sim$2 s window of past keys and values, enabling efficient reuse of context.

**Learnable bottleneck with Factorized VQ.** To remove residual speaker information and regularize the content space, we used a factorized vector-quantized (VQ) bottleneck (Ju et al., 2024). The 512-dim encoder output is first compressed to an 8-dim latent vector via a learned projection, quantized using a learnable codebook of size 4096, and then projected back to 512 dimensions. This compress-then-discretize design encourages the model to learn discrete, speaker-independent units while preserving linguistic fidelity for downstream synthesis.

**Training objective.** The encoder and bottleneck are optimized with a cross-entropy objective against discrete pseudo-labels obtained by applying $k$-means clustering ($N = 200$ centroids) to the 9th layer activations of a HuBERT-base[4]. The encoder, projection layers, and VQ codebook are trained jointly so that the bottleneck learns to predict discrete units, while the CNN encoder and MHSA context layer capture both local phonetic cues and longer-range temporal dependencies. This stage is fully self-supervised and does not require transcripts or text alignments.

## 3.2 TIME-VARYING TIMBRE (TVT) REPRESENTATION

Conventional VC systems represent speaker identity with a single global embedding $g \in \mathbb{R}^d$, extracted from a reference utterance or corpus. While $g$ captures a speaker's timbre at a coarse level, it is a static vector. In contrast, content embeddings $\{c_t\}_{t=1}^T$ vary at the frame level, encoding phonetic and prosodic dynamics. This mismatch between static speaker and dynamic content representations often produces over-smoothed timbre, limited expressivity, or degraded consistency under challenging conditions. To overcome this, we introduce a *time-varying timbre representation* that allows the speaker embedding to evolve in sync with the content –see Figure 2a.

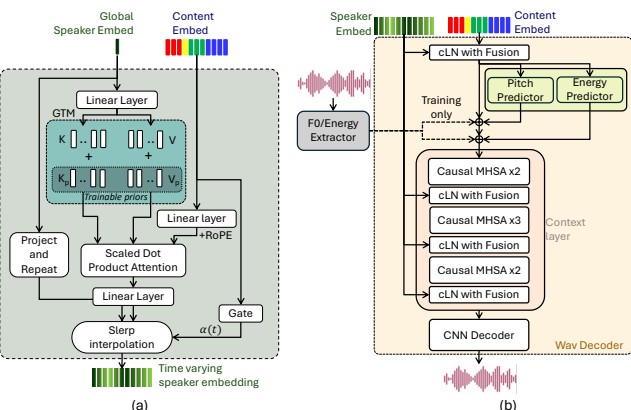

Figure 2: Architecture details for (a) TVT processing block, (b) waveform decoder.

**Global Timbre Memory.** To obtain a global speaker embedding, we concatenate two complementary embeddings–noise-robust X-vectors (Snyder et al., 2018) and context-aware ECAPA-TDNN (Desplanques et al., 2020), which improves downstream anonymization quality (Meyer et al., 2022). We project this global embedding $g$ into a *Global Timbre Memory* (GTM), parameterized as $K$ key–value pairs $\{(k_i, v_i)\}_{i=1}^K$, with $k_i, v_i \in \mathbb{R}^d$. The GTM employs a dual representation: a speaker-specific component generated by an MLP from $g$, and learnable prototype parameters $k_i^{\text{prior}}, v_i^{\text{prior}}$

---

[4]https://github.com/facebookresearch/fairseq

shared across all speakers. Formally, each key-value pair is computed as:

$$k_i = \text{MLP}_k(g)_i + k_i^{\text{prior}}, \quad v_i = \text{MLP}_v(g)_i + v_i^{\text{prior}}, \tag{1}$$

where the priors act as universal timbre prototypes that capture phoneme-agnostic characteristics (e.g., breathiness, nasality) common across speakers, while the MLP output modulates these prototypes to reflect individual voice identity. This design provides a strong inductive bias, improving sample efficiency and training stability, particularly in low-data regimes or with unseen speakers. Intuitively, the GTM decomposes the timbre into multiple "facets" (e.g., spectral color, nasality, brightness), each stored in a slot. At each time step $t$, the content embedding $c_t$ attends over the keys to retrieve a weighted timbre component: $v_t = \text{Attn}(c_t, \{k_i\}, \{v_i\})$, where scaled dot-product attention produces a distribution over slots. This enables the model to select the most relevant timbre sub-components given the current phonetic and/or prosodic context.

**Gating and Interpolation.** To balance stability with flexibility, a gating network computes a scalar $\alpha_t \in [0, 1]$ that modulates how much the embedding should deviate from the global timbre. The final time-varying embedding $s_t$ is obtained by interpolating between $g$ and $v_t$: $s_t = \text{Slerp}(g, v_t; \alpha_t)$, where Slerp denotes spherical linear interpolation, which respects the hyperspherical geometry of the embedding space, ensuring smooth trajectories and preserving angular distances. Specifically, Slerp interpolates along the geodesic (great circle arc) connecting two points on the unit hypersphere, maintaining constant angular velocity: $\theta_t = (1 - \alpha_t)\theta_g + \alpha_t\theta_v$ (Shoemake, 1985). This geometric property has been shown to better preserve identity characteristics in high-dimensional embedding spaces compared to Euclidean interpolation (Zhong et al., 2025). This avoids unnatural distortions that could arise from linear interpolation in Euclidean space, which creates shortcuts through the interior of the hypersphere, requiring re-normalization that distorts angular relationships and can cause perceptual artifacts.

**Intuition.** Conceptually, $g$ provides a stable "base palette" for a speaker's identity, while the GTM decomposes this palette into finer brushes. The learnable prototypes serve as a universal "brush set" that is refined during training to capture common timbre characteristics across all speakers, while the speaker-specific MLP output adjusts the bristle texture and pressure to match individual voices. At each frame, the content embedding chooses which brushes to apply, adjusting timbre in context-sensitive ways. Gating mechanism controls how bold these adjustments are, and Slerp guarantees that the blend remains smooth and consistent. The resulting sequence of TVT embeddings $\{s_t\}$ preserves global identity while adapting locally, enabling more natural and controllable synthesis.

### 3.3 STREAMING WAVEFORM DECODER

**Speaker Conditioning.** We condition the encoder output and subsequent latent representations in the decoder on TVT embeddings –see Figure 2b. Timbre conditioning is achieved through a *Conditional Layer Normalization with Fusion* module. Given latent features $x \in \mathbb{R}^{B \times 512 \times T}$ and time-varying speaker embeddings $s \in \mathbb{R}^{B \times 192 \times T}$, the module normalizes $x$, then generates per-frame scale and shift coefficients $(\gamma, \beta)$ from $s$. The re-normalized features are fused with a gated, normalized version of $s$ and projected back to the latent dimension:

$$y_t = \text{Proj}\Big((1 + \gamma_t) \cdot \text{Norm}(x_t) + \beta_t \,\|\, g_t \cdot \text{Norm}(s_t)\Big),$$

where $g_t$ is a learned gate and $\|$ denotes concatenation. This design allows the model to integrate dynamic speaker information while providing stability to the normalized content features.

**F0/Energy Predictor.** Prosodic variation is introduced by lightweight predictors for fundamental frequency ($F0$) and energy. Each predictor is a 2-layer causal CNN (kernel=3) with ReLU activations and a final point-wise projection. During training, these modules are supervised with ground-truth $F0$ and energy extracted from the waveform; at inference, their predictions are injected into the feature stream, enabling explicit control over pitch and loudness.

**Decoder.** The decoder mirrors the content network and reconstructs waveforms from the conditioned embeddings. It begins with a *context layer* composed of a stack of 8 causal MHSA blocks with a fixed look-back window of $W = 2\,\text{s}$ but *no future peeking*. A ring KV cache maintains this rolling past context for efficient streaming. Following the context layer, a *CNN decoder* inverts the encoder's temporal compression via four causal ConvTranspose1D stages with strides $[2, 4, 5, 8]$ (the reverse of the encoder's downsampling). These stages progressively upsample the sequence from $\approx 50\,\text{Hz}$

back to $16\,\mathrm{kHz}$ (overall factor $\prod s_i = 2{\times}4{\times}5{\times}8 = 320$). Each upsampling stage is interleaved with a lightweight residual block structurally matched to the encoder (two 1-D conv with kernels 3 and 1, dilation base 2, ELU activations, and a true skip connection), preserving fine-scale detail contributed by content, time-varying timbre, and prosody streams.

**Training objective.** The decoder is optimized with a multi-objective loss that balances spectral fidelity and naturalness. We combine an L1 reconstruction loss $\mathcal{L}_{\mathrm{mel}}$ on log-Mels at multiple window lengths (2-128 ms) with adversarial objectives from multi-period waveform and multi-band spectrogram discriminators, $\mathcal{L}_{\mathrm{adv}}$, a feature-matching term $\mathcal{L}_{\mathrm{fm}}$ on discriminator activations (Kumar et al., 2023), and F0/energy predictor L2 loss $\mathcal{L}_{\mathrm{fo\text{-}e}}$. The total loss is

$$\mathcal{L}_{\mathrm{total}} = \lambda_{\mathrm{mel}}\,\mathcal{L}_{\mathrm{mel}} + \lambda_{\mathrm{adv}}\,\mathcal{L}_{\mathrm{adv}} + \lambda_{\mathrm{fm}}\,\mathcal{L}_{\mathrm{fm}} + \lambda_{\mathrm{f0\text{-}e}}\,\mathcal{L}_{\mathrm{f0\text{-}e}},$$

where $\lambda_{\{\cdot\}}$ weight the respective terms. We use $\lambda_{\mathrm{mel}} = \lambda_{\mathrm{f0\text{-}e}} = 20$, $\lambda_{\mathrm{adv}} = 1$ and $\lambda_{\mathrm{fm}} = 2$.

## 4 EXPERIMENTAL SETTING

**Datasets.** We train our content encoder and decoder on the LibriTTS corpus (Zen et al., 2019), which provides roughly 600 hours of read English speech. Pretrained speaker encoders are taken from SpeechBrain[5] (Ravanelli et al., 2021), trained on the VoxCeleb corpus (Nagrani et al., 2017).

**Tasks.** We consider two tasks for evaluation: *voice conversion (VC)* and *speaker anonymization (SA)*. For VC, we use CMU ARCTIC (Kominek, 2003), L2-ARCTIC (Zhao et al., 2018), and VCTK(Veaux et al., 2017) as source datasets, and EMIME (Wester, 2010) as the target. Specifically, for each speaker we randomly select 50 utterances and convert them into a different set of random target utterances from the EMIME English subset. For SA, we follow the VPC 2024 evaluation protocol[6]. Namely, we use the LibriSpeech dev-clean and test-clean subsets (Panayotov et al., 2015) to compute objective intelligibility and privacy metrics. Anonymized samples are generated by randomly selecting a target utterance from EMIME.

**Metrics.** For the VC task, we report two performance metrics: synthesis quality and speaker similarity. To measure synthesis quality we use NISQA-MOS (Mittag et al., 2021), a model that predicts human-ratings of mean opinion scores (MOS) for quality and naturalness. To quantify speaker similarity we use the cosine similarity between speaker embeddings on concatenated X-vectors and ECAPA-TDNN embeddings. For the anonymization task, we use EER as a measure of privacy and WER as a measure of intelligibility, per the VPC protocol. To evaluate streaming performance, we measure latency on an NVIDIA RTX 500 Ada GPU and a dual-socket AMD EPYC 7543 CPU.

**Model training.** All modules are trained using the AdamW optimizer with an initial learning rate of $5 \times 10^{-4}$ and a batch size of 16 (random 3-sec clips). The content encoder was optimized with a `ReduceLROnPlateau` learning rate scheduler, whereas the waveform decoder employed the `ExponentialLR` scheduler with decay factor $\gamma = 0.999996$. The encoder and waveform decoder comprised $37.5\,\mathrm{M}$ and $48.7\,\mathrm{M}$ trainable parameters, respectively. Both content encoder and the waveform decoder were trained independently for $500k$ steps on an NVIDIA RTX 5000 Ada GPU.

**Baseline systems.** We compare TVTSyn against four SOTA streaming methods: SLT24 (Quamer & Gutierrez-Osuna, 2024), DarkStream (DS) (Quamer & Gutierrez-Osuna, 2025a), and GenVC (Cai et al., 2025). SLT24 is fully causal CNN based archiecture and uses bottleneck features as content embedding. DS is a CNN-transformer based system that uses a 140ms lookahead and applies $k$-means clustering to the content embeddings. For GenVC, a LM based generative model, we evaluated two configurations: GenVC-small and GenVC-large under the streaming decoding protocol, with the top-$k$ parameter set to 1. It should be noted that the GenVC encoder is non-causal (i.e., it generates content features after consuming the entire source utterances), so it only streams at the decoder level. Thus, the two GenVC baselines are at an advantage when compared to the rest of the models (SLT24, DS, TVTSyn), which operate in a streaming fashion end-to-end, and therefore are affected by errors (i.e., mis-predicted future tokens) in the causal encoder. For our proposed TVTSyn model, we evaluated the full configuration shown in Figure 1 (P), along with three ablations: (-VQ) removing the compression–VQ step, (-TVT) removing the TVT processing block, and (-VQ/-TVT) removing both.

---

[5]https://huggingface.co/speechbrain
[6]https://github.com/Voice-Privacy-Challenge/Voice-Privacy-Challenge-2024

## 5 RESULTS

### 5.1 CONTENT REPRESENTATION

Figure 3 shows t-SNE visualizations of the content embeddings, averaged across time at the utterance level. Panel (a) shows the continuous embeddings at the output of the content encoder, colorcoded by speaker, markers indicating native vs. non-native speakers. We observe two broad clusters, separating native and non-native speakers, with tight sub-clusters for each individual speaker. This indicates that the continuous content embeddings retain significant speaker cues. Panel (b) shows the t-SNE plot of the logits representation, obtained by linearly projecting the continuous embeddings onto the codebook. While the logits representation still separates native from non-native speakers, the within-speaker clusters are noticeably looser compared to the continuous embeddings. Panels (c) and (d) show t-SNE plots for the bottleneck and VQ bottleneck representations, respectively. In both cases, within-speaker clustering is further reduced compared to the logits embeddings, indicating that both bottleneck approaches substantially reduce speaker leakage. Though native vs. non-native clusters are still separable in (c) and (d), this reflects the fact that non-native accents do alter the content of speech, both at the segmental level (e.g., phonetic substitutions) and the prosodic level (e.g., stress-timing in English vs. syllable-timing in Spanish and Mandarin).

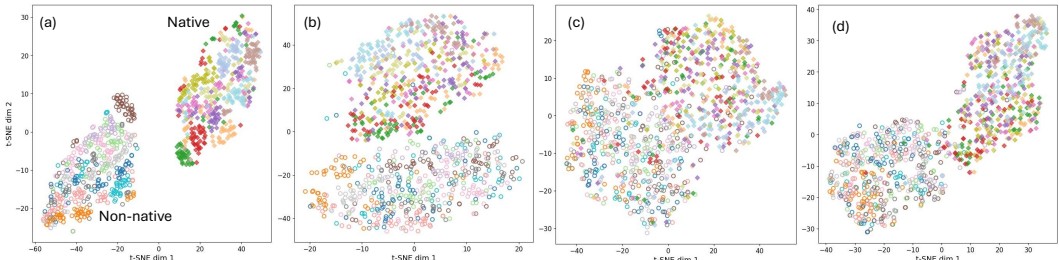

Figure 3: t-SNE visualization of content embeddings, color-coded by speaker. Markers denote native (♦) or non-native (○). (a) Continuous embeddings, (b) logits, (c) bottleneck, and (d) VQ bottleneck.

### 5.2 TIME VARYING TIMBRE REPRESENTATION

Figure 4 illustrates how the speaker-processing block yields a content-synchronous, time-varying timbre. The content-GTM attention heatmap in panel (a) shows sparse islands and horizontal bands across tokens, indicating that different timbre facets are retrieved at different moments while a few facets are reused across longer spans. The thin Top-1 raster (panel b) highlights discrete facet switching aligned with phonetic/prosodic transitions. Panel (c) shows PCA trajectories for pre-Slerp embedding $v_t$ and final embedding $s_t$ along with the global timbre $g$. Here the $v_t$ wander broadly, whereas $s_t$ forms a compact, smooth tube around the global point. This shows that the interpolation path keeps identity geometry intact while allowing local movement. Finally, the GTM usage and geometry (panels d and e) show a dispersed memory with non-collapsed token usage, suggesting that the model learned a diverse set of reusable timbre facets.

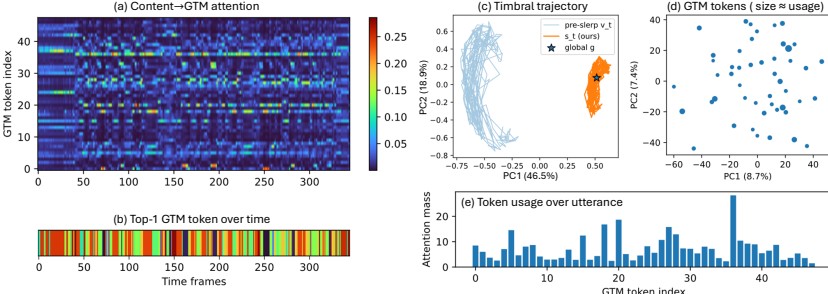

Figure 4: Qualitative analysis of time-varying timbre for the text: *"Six spoons of fresh snow peas, five thick slabs of blue cheese, and maybe a snack for her brother Bob."*. (a) Content-GTM attention map with (b) Top-1 strip shows content-dependent selection of timbre facets, (c) PCA trajectories (pre-slerp vs. final), (d) PCA projection of GTM value tokens (size ∝ usage) and (e) token-usage histogram indicate diverse, non-collapsed facets.

## 5.3 SPEAKER TRANSFER (VOICE CONVERSION)

Figure 5 summarizes objective evaluation of VC performance across baselines (SLT24, DS, GenVC), the proposed model (P), and ablations (-VQ, -TVT, -VQ/-TVT). The proposed model achieves the strongest anonymization performance of all models (i.e., lowest Src-SIM and highest Trg-SIM), and the second highest NISQA scores, marginally lower than baseline SLT24 (4.01 vs. 3.91). As a reference, Figure 5 also includes scores for unmodified source speech (NISQA: 4.41). Note that the proposed model achieves the same Trg-SIM score (0.77) as that of within-speaker comparisons in real speech (i.e., Src-SIM for the reference source utterances), and similar Src-SIM (0.48) as that of between-speaker comparisons in real speech (i.e., Trg-SIM for the reference source utterances, 0.48). *In other words, our proposed model generates speech that is as similar to the target speaker as any two real utterances from that target speaker, and as dissimilar from the source speaker as that between the source speaker and any other speaker.*

The ablation study show that removing the TVT of VQ steps lead to a significant reduction in NISQA scores (from 3.91 to 3.42/3.44), and that removal of both TVT and VQ steps degrades NISQA scores even further (3.179). However, neither TVT or VQ appear to play a major role in anonymization performance. Finally, the two GenVC baselines achieve the weakest anonymization performance, with Trg-SIM scores far lower than those of the remaining models, and also modest NISQA scores despite GenVC using a non-causal content encoder.

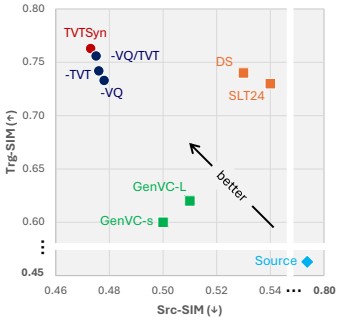 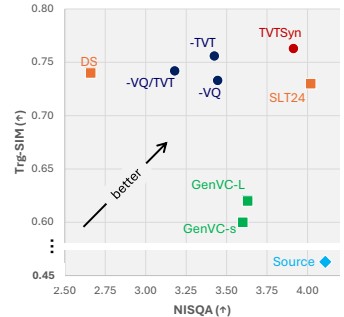

Figure 5: Objective evaluation results for voice conversion. Src-SIM: cosine similarity b/w VC and source speaker; Trg-SIM: cosine similarity b/w VC and target speaker; NISQA-MOS: Speech Quality and Naturalness Assessment. Src-SIM and Trg-SIM for source speech (i.e., unaltered) reflect within- and between-speaker similarity, respectively.

Table 1 shows the contribution of each component in the TVT processing block. Notably, Src-SIM remains constant at 0.48 across all ablations, indicating that privacy is determined by the overall architecture rather than the fine-grained TVT design and that TVT's primary role is preserving synthesis quality while maintaining anonymization. We observe that removing GTM produces the largest drop in acoustic quality (3.91 to 3.45), indicating that the content-synchronous timbre is essential for naturalness. Removing learnable priors also causes a notable degradation (3.91 to 3.62), as the model loses universal timbre bases that enable efficient generalization. Replacing Slerp with linear interpolation and replacing gating with fixed $\alpha = 0.5$ produce modest but measurable degradations, validating these design choices. The same is observed when reducing GTM capacity from 48 to 24 tokens or 12 tokens, confirming our choice of 48 tokens balances expressivity and efficiency.

Table 1: Ablation study of components in the TVT speaker processing block. All ablations maintain similar privacy, while synthesis quality degrades without key components.

| | Models | Src-SIM ($\downarrow$) | Tgt-SIM ($\uparrow$) | NISQA MOS ($\uparrow$) |
|---|---|---|---|---|
| **Proposed** | TVTSyn (48 GTM tokens) | **0.47** | **0.77** | **3.91** |
| **TVT ablations** | w/o gating | 0.48 | 0.76 | 3.80 |
| | w/o slerp | 0.48 | 0.76 | 3.75 |
| | w/o GTM | 0.48 | 0.75 | 3.45 |
| | w/o prior | 0.48 | 0.75 | 3.62 |
| | w/ 24 GTM tokens | 0.48 | 0.76 | 3.81 |
| | w/ 12 GTM tokens | 0.48 | 0.75 | 3.73 |

We corroborated the objective results in Figure 5 with perceptual listening tests on Amazon Mechanical Turk (N=20)[7].To evaluated speaker transfer, we used a standard ABX protocol, where participants had to select whether utterance X (voice conversion) sounded closer to A or B (source and target speaker, counterbalanced), and also reported their confidence on a 1–7 scale (7 = extremely confident; 1: not confident at all). For speech quality, listeners provided mean-opinion scores for individual utterances. Results are summarized in Table 2. The proposed system achieved the highest MOS across all models, and only marginally below the perceived speech quality of unedited source utterances. For speaker transfer, the proposed system also achieved the highest verifiability rate (74.33%) of all models with a high confidence rate (5.02).

Table 2: Human listening test (N=20). Mean opinion score (MOS) for audio quality (95% confidence interval) and speaker verifiability scores.

|  | Source | SLT24 | DS | GenVC-s | TVTSyn |
|---|---|---|---|---|---|
| **MOS** | $3.84 \pm 0.10$ | $3.77 \pm 0.09$ | $3.49 \pm 0.13$ | $3.63 \pm 0.11$ | $\mathbf{3.82 \pm 0.10}$ |
| **Prefer Target Speaker** | - | 68.00% | 69.33% | 70.67% | **74.33%** |
| **Avg. Confidence Rating** | - | 5.06 | 4.99 | 5.04 | 5.02 |

## 5.4 Speaker Anonymization

Following the VPC'24 protocol (Tomashenko et al., 2024), we computed EERs under two attacker models: a lazy-informed attacker (knows the algorithm but has no enrollment data) and a stronger semi-informed attacker (retrains ASV models using anonymized enrollment). Higher EER indicates better anonymization. We also report Word Error Rate (WER) for intelligibility and Unweighted Average Recall (UAR) for emotion characteristics. TVTSyn achieves favorable privacy–utility balance: strong anonymization (EER = 47.6% lazy, 14.6% semi), excellent intelligibility (WER = 5.35%), and intentional emotion suppression (UAR = 37.32%) for enhanced privacy—see Table 3. Notably, TVTSyn outperforms all streaming baselines on utility (WER: 5.35% vs. SLT24's 5.70%, DarkStream's 10.80%) while maintaining competitive privacy.

**Comparison with VPC'24 Offline Systems.** Table 3 includes VPC'24 baselines (B2-B6) and top participants (T10-C3, T9, T8-4, T38-M1) for context; however, direct comparison is problematic due to fundamentally different constraints. VPC systems operate offline with full-utterance bidirectional context and diverse pseudo-speaker generation (*e.g.*, GAN-based sampling), whereas TVTSyn operates causally with <80ms latency using 28 fixed pseudo-speakers–a simplification for this study; future work will incorporate pseudo-speaker generation. Additionally, design goals differ: VPC participants optimize for emotion preservation (UAR: 60–65%), whereas TVTSyn targets emotion suppression (UAR = 37.32%), reflecting comprehensive identity masking including paralinguistic traits. For our privacy-first objective, lower UAR is desirable.

Our contribution is not surpassing offline systems, but demonstrating that streaming anonymization under strict latency constraints can maintain strong privacy and utility—addressing real-world deployment needs (teleconferencing, live translation) where offline processing is infeasible.

Table 3: VPC'24 evaluation: WER, EER and UAR as measure of intelligibility, anonymization strength and emotion preservation respectively.

|  | Models | WER ($\downarrow$) | EER (lazy-informed, $\uparrow$) | EER (semi-informed, $\uparrow$) | UAR (emotion) |
|---|---|---|---|---|---|
| **Baselines** | SLT24 | 5.70 | 31.40 | 10.12 | 57.00 |
|  | DS | 10.80 | **49.09** | 20.83 | 34.49 |
|  | GenVC-s | 8.20 | 48.48 | 15.94 | 34.23 |
| **VPC24 Baselines** | B2 | 10.20 | - | 5.99 | 54.55 |
|  | B3 | 4.35 | - | 26.28 | 37.83 |
|  | B4 | 6.02 | - | 31.49 | 42.37 |
|  | B5 | 4.55 | - | 34.35 | 38.12 |
|  | B6 | 9.39 | - | 22.09 | 36.26 |
| **VPC24 Participants** | T10-C3 (Yao et al., 2024) | 2.62 | - | 37.34 | 65.23 |
|  | T9 (Tan et al., 2024) | **2.35** | - | 34.27 | 60.82 |
|  | T8-4 (Xinyuan et al., 2024) | 3.75 | - | **48.25** | 30.35 |
|  | T38-M1 (Le Blouch et al.) | 8.31 | - | 33.31 | 32.23 |
| **Proposed** | TVTSyn | 5.35 | 47.55 | 14.57 | 37.32 |

[7]We acknowledge that large-scale listening tests may still be required for comprehensive validation.

## 5.5 REAL-TIME PERFORMANCE

We report synthesis latency and real-time factor (RTF) for all models using chunk sizes of 60 ms and 100 ms. As shown in Table 4, TVTSyn achieves latencies of $\approx$79 ms on GPU and $\approx$132 ms on CPU, with RTFs of 0.31 and 1.20, respectively, comfortably within real-time bounds. Compared to SLT24 and DS, our models reduce both latency and RTF while maintaining comparable quality. Importantly, DS requires a 140 ms lookahead at the encoder, effectively raising end-to-end latency well above the reported figure. In contrast, our models operate fully causally at runtime without lookahead, making them substantially more suitable for low-latency deployment. While results are reported for 60 ms and 100 ms chunk size, the architecture scales robustly to smaller or larger chunks with consistent real-time performance, ensuring flexibility across diverse streaming scenarios. We did not include real-time results for GenVC, since its encoder is non-causal and streams only at the decoder level, making latency comparisons with fully causal systems unfair.

Table 4: Latency and RTF on CPU and GPU for proposed and streaming baselines.

| Model | Chunk size = 60 ms | | | | Chunk size = 100 ms | | | |
| | CPU | | GPU | | CPU | | GPU | |
| | Latency (ms) | RTF | Latency (ms) | RTF | Latency (ms) | RTF | Latency (ms) | RTF |
|---|---|---|---|---|---|---|---|---|
| SLT24 | 187.11 | 2.119 | 86.49 | 0.441 | 244.31 | 1.443 | 123.55 | 0.236 |
| DarkStream (DS) | 127.02 | 1.117 | 76.12 | 0.269 | 172.45 | 0.724 | 119.12 | 0.191 |
| TVTSyn | 131.76 | 1.196 | 78.51 | 0.308 | 186.16 | 0.862 | 119.77 | 0.198 |

## 6 DISCUSSION

We proposed TVTSyn, a streaming model for voice conversion and speaker anonymization that uses a time-varying representation of speaker timbre to match the temporal granularity of linguistic content, enabling a better trade-off between naturalness, speaker similarity, and privacy under strict latency constraints. Our results show that the privacy–utility balance is fundamentally architectural: by resolving the mismatch between static speaker embeddings and dynamic content, TVT preserves expressivity without weakening anonymization. The factorized VQ bottleneck further regularizes content, improving intelligibility while maintaining anonymizaiton performance. Ablations reinforce this view: removing either TVT or VQ degrades quality and VC performance. Together, these findings show that controlling the interaction between timbre variability and VQ strength is the key to managing the privacy–utility trade-off, with ablation models allowing control of utility or privacy depending on deployment needs.

Future work will make this alignment more structured and controllable. One direction is to explicitly disentangle static traits (e.g., accent, age, sex) from dynamic attributes(e.g., emotion, speaking style). We envision a two-path design: a "static" pathway to modify age/sex, and a frame-synchronous "style" pathway driven by TVT. This setup would enable controllable anonymization, e.g., selectively masking emotion or sex cues while preserving intelligibility. A second direction is to extend the Global Timbre Memory to include timbre+style facets, exposing simple controls for emotion and expressivity. In this setting, anonymization could be combined with behavioral camouflage strategies–prosody shaping, hesitations, or filler words–that break habitual patterns without increasing WERs. A third direction is to broaden the scope to cross-lingual and code-switching speech, where accent interacts with phonotactic variation; here, a lightweight language-ID head could help stabilize TVT across language switches. We will explore adaptive chunking and hierarchical KV caches to further reduce latency, alongside edge-friendly deployment methods such as quantization and low-rank adapters, enabling robust CPU-only real-time operation. Finally, robustness to noisy and reverberant acoustic environments can be enhanced through data augmentation techniques during training, such as additive noise, reverberation simulation, and codec artifacts, as demonstrated in recent work on noise-robust speech synthesis (Ranjan et al., 2024; Lakshminarayana et al., 2025), as systematic evaluation under acoustic degradation remains an important direction for real-world deployment.

### ACKNOWLEDGMENTS

Supported by the Intelligence Advanced Research Projects Activity (IARPA) via Department of Interior/Interior Business Center (DOI/IBC) contract number 140D0424C0066. The U.S. Government

is authorized to reproduce and distribute reprints for Governmental purposes notwithstanding any copyright annotation thereon. Disclaimer: The views and conclusions contained herein are those of the authors and should not be interpreted as necessarily representing the official policies or endorsements, either expressed or implied, of IARPA, DOI/IBC, or the U.S. Government.

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

## A ARCHITECTURE DETAILS

### A.1 MODEL CONFIGURATIONS

Our system comprises four main modules: (i) a causal *content encoder* with a factorized VQ bottleneck, (ii) a *speaker processing block* that produces time-varying timbre (TVT) via a Global Timbre Memory (GTM) with gate and Slerp, (iii) a causal *context layer* (MHSA) for frame-level conditioning, and (iv) a causal *waveform decoder* (SEANet). The high-level dataflow matches the training and inference diagrams (content/VQ, GTM+gate+Slerp, and cLN with Fusion) shown in Fig. 1 and 2.[8]

**Signal scale and rates.** Audio is 16 kHz; frames are 50 Hz (20 ms hop). All intermediate features (content, TVT, prosody) are aligned at the frame clock.

**Content encoder.** A lightweight causal SEANet stack maps audio to 512-D frame embeddings using strides $[8, 5, 4, 2]$ (overall $\times 320$) with ELU activations, kernel sizes $7/3/3$, dilation base 2, and true-skip connections. A causal 8-layer MHSA context block ($d_{model}$=512, 8 heads, RoPE positional encoding, FFN 2048) provides a 2 s look-back (`context`=100 frames). During training only, a masked right context of 4 frames ($\approx$ 80 ms) is enabled to stabilize learning; at inference, *right context is limited by chunk-size, i.e., applies within chunk* (chunk-wise causal).

**Factorized VQ bottleneck.** The 512-D content embedding is projected to an 8-D latent and quantized with a 4096-entry codebook ($codebook\_dim = 8$, $codebook\_size = 4096$); commitment loss 0.15 with L2 code normalization. The quantized latent is then projected back to 512-D and passed forward. This reduces speaker leakage in content while preserving lexical information.

**Speaker processing (TVT).** Following the diagram, a global speaker vector is expanded to a GTM; content frames attend ($attention\_dim = 128$) to GTM to retrieve a facet, a gate $\alpha(t) \in [0, 1]$ modulates deviation from the global vector, and $\text{Slerp}\big(\text{global}, \text{facet}; \alpha(t)\big)$ yields the time-varying timbre embedding. These TVT features condition decoding via cLN with Fusion.[9] The TVT interface dimensions follow the config ($global\_timbre\_dimension = 704$, $timbre\_cond\_dim = 192$).

**Context layer for synthesis.** A causal frame-rate synthesizer (2 s look-back, no right context) refines 512-D content features and fuses TVT and prosody. It uses 8-head MHSA with RoPE, FFN 2048.

**Waveform decoder.** A causal SEANet vocoder mirrors the encoder's strides $[2, 4, 5, 8]$ back to waveform, using ELU activations, dilation base 2, and weight normalization. Conditioning is injected via *cLN with Fusion* at multiple stages (global content normalization plus affine modulation from TVT and prosody).

---

[8]See the training and detailed TVT/conditioning diagram for reference.

[9]TVT attention, gate, and Slerp; and cLN with Fusion are illustrated in Figure 2

Table 5: Core hyperparameters.

| Item | Setting |
|------|---------|
| Sample / frame rate | 16,000 Hz / 50 Hz (20 ms hop) |
| Content channels / dim | $d$=512 |
| SEANet strides | Encoder [8, 5, 4, 2]; Decoder [2, 4, 5, 8] |
| Activations / norm | ELU; encoder `norm=weight_norm`, decoder `weight_norm` |
| Transformer (enc) | 8 layers, 8 heads, $d\_model$=512, FFN 2048, RoPE, layer_scale=0.01 |
| Context window | 100 frames (2 s) look-back; *right_context=4* |
| Transformer (frame synth) | causal, 8 heads, $d\_model$=512, FFN 2048, RoPE, *right_context=0* |
| VQ bottleneck | $|\mathcal{C}|$=4096, code dim 8, commitment 0.15, L2 normalize |
| TVT / fusion dims | *timbre_dimension*=704, *timbre_cond_dim*=192 *attention_dim*=192 |
| Prosody | F0/Energy predictors (causal), fused via cLN with Fusion |

## A.2 STREAMING IMPLEMENTATION

The system is designed for causal, low-latency streaming synthesis. All modules are adapted to operate on fixed-length chunks with persistent states across calls.

**Causal Convolutions.** Convolutions in the encoder and decoder are wrapped with ring-buffer state management. At each step, only the newest samples are convolved, while cached activations from previous chunks are reused. This avoids redundant computation and ensures causality.

**Streaming Attention.** Transformer blocks maintain a rolling key–value cache. During inference, only the most recent 2 seconds of context are retained, with a limited 4-frame future peek ($\sim$80 ms). The cache is updated incrementally per chunk, allowing efficient streaming inference without reprocessing past context.

**Prosody Predictors.** Pitch and energy predictors are causal CNNs that operate chunk-wise. Their states are maintained so that predictions are consistent across chunk boundaries.

Table 6 summarizes the main streaming parameters.

| Parameter | Value |
|-----------|-------|
| Chunk size | 60 ms (default, tested 20–140 ms) |
| Encoder buffer | 320-sample hop, causal ring buffer |
| Attention context | 2 s look-back, 80 ms look-ahead |
| Decoder overlap-add | 20 ms overlap |
| Streaming states | Encoder, Transformer KV cache, Decoder buffers |

Table 6: Summary of streaming implementation.

## B EVALUATION

### B.1 LATENCY MEASUREMENTS

Latency is defined as the sum of the chunk size and the processing time per chunk (in milliseconds), averaged over all chunks within an utterance. The real-time factor (RTF) is computed as the ratio of processing time to chunk size for each chunk, then averaged across all chunks of the same utterance. To avoid initialization overhead, we first performed a warm-up on 10 utterances. Latency and RTF were then measured on 100 utterances, and we report the mean values across samples.

### B.2 PERCEPTUAL LISTENING TESTS

We conducted two subjective evaluations with 20 unique participants for each test. All listeners were based in the United States at the time of recruitment. The study protocol was approved by the Institutional Review Board of Texas A&M University and deployed through Amazon Mechanical Turk.

### B.2.1 MEAN OPINION SCORE (MOS)

To assess perceived quality, participants rated individual utterances on a five-point mean opinion score (MOS) scale, where higher ratings correspond to more natural speech and lower distortion. The scale definitions are provided in Table 7. Prior to evaluation, listeners were given example clips representing each score to help them calibrate their judgments Loizou (2011). These calibration samples were drawn from the 2018 Voice Conversion Challenge dataset Lorenzo-Trueba et al. (2018). Each participant evaluated 15 utterances per system.

Table 7: Mean Opinion Score rating scale.

| Rating | Speech Quality | Distortion Level |
|--------|---------------|------------------|
| 5 | Excellent | Imperceptible |
| 4 | Good | Slight but not distracting |
| 3 | Fair | Noticeable but tolerable |
| 2 | Poor | Distracting but intelligible |
| 1 | Bad | Very distracting or unnatural |

### B.2.2 SPEAKER VERIFIABILITY TEST

To measure speaker similarity, we conducted an ABX test. Each trial presented three recordings (X, A, B). Listeners judged whether A or B sounded more like X, then rated their confidence on a seven-point scale (7: extremely confident; 5: quite a bit confident; 3: somewhat confident; 1: not confident at all). Each participant evaluated 15 randomly sampled voice-converted utterances from every system. To ensure judgments reflected voice characteristics rather than lexical overlap, the source and target recordings (A/B) contained different content than the converted sample.

## C LLM USAGE

Large language models were employed only to assist with writing tasks, such as improving clarity, grammar, and style. No experimental design, data analysis, or result interpretation relied on automated tools.

