# OpenReview forum: "TVTSyn: Content-Synchronous Time-Varying Timbre for Streaming Voice Conversion and Anonymization"
_ICLR.cc/2026/Conference — ICLR 2026 Poster_

### Official Review · Reviewer_1kxu · 2025-10-31

**Soundness:** 1
**Presentation:** 2
**Contribution:** 2
**Rating:** 2
**Confidence:** 5

**Summary:**

This paper state the static-dynamic mismatch between fixed speaker embeddings and time-varying linguistic content. The proposed TVTSyn model introduces a content-synchronized time-varying timbre representation, complemented by a factorized VQ bottleneck, to balance low latency, naturalness, and privacy.

**Strengths:**

This paper is well-structured and provide analysis on the effcient of the proposed system.

**Weaknesses:**

- The paper’s central claim resolving the static-dynamic mismatch via time-varying timbre is not sufficiently novel. Prior work has already explored dynamic speaker conditioning for speech synthesis.
- The dataset employed in experiments is not convince and popular, which make me confuse the correctness of the conclusion.
- Incomplete baseline comparisons with voice privacy challenge baseline systems or other popular speaker anonymization systems.

**Questions:**

See weaknesses

---

> ### Author Response · Authors · 2025-11-21
> **We appreciate Reviewer 1kxu for the feedback. Part of the criticism is warranted since we had failed to specify our contributions. We have made substantial revisions to the manuscript to clarify our novelty and experimental rigor.**
>
> We appreciate Reviewer 1kxu for the feedback. Part of the criticism is warranted since we had failed to specify our contributions. We have made substantial revisions to the manuscript to clarify our novelty and experimental rigor. As the reviewer suggested, we have also added new VPC baseline comparisons. However, we respectfully disagree that the datasets are not convincing/popular, but would be happy to consider other datasets the reviewer thinks may be more suitable for our work.
>
> Q1: Insufficient novelty—prior work explores dynamic speaker conditioning.
> - Our novelty lies in the combination of learnable prototypes + causal streaming + geometric interpolation for voice anonymization—no prior work achieves all three.  More specifically:
>      - Causal streaming architecture: Unlike GenVC/DAFMSVC (non-causal, offline), TVTSyn operates causally with <80ms latency. This fundamentally changes the problem—causal systems cannot access future context, requiring windowed attention, RoPE positional encoding, and careful content-speaker disentanglement using factorized VQ. Prior non-causal methods leverage full-utterance context; we cannot. Section 2 (Related Work) now explicitly contrasts causal vs. non-causal implications.
>      - Learnable universal prototypes: Our GTM employs shared prototype parameters capturing universal timbre characteristics across speakers (Equation 1). This differs from FreeVC (static embeddings), GenVC (instance-specific queries), and DAFMSVC (no shared prototypes). Ablation (Table 1) shows removing prototypes causes -7.4% NISQA degradation.
>      - Spherical interpolation: Slerp preserves embedding geometry, validated through ablation (-4.1% NISQA). See Section 3.2 for the justification of design choice.
> - We have added these three key distinctions to Sections 2 and 3.2 of the revised manuscript (see highlights).
>
> Q2: Dataset not convincing/popular.
> - We follow the VoicePrivacy Challenge 2024 protocol (Section 5), which represents the community consensus. Datasets include LibriTTS (585 speakers, used in 1,300+ publications), VCTK (109 speakers, industry standard), and L2-ARCTIC (24 non-native speakers, an established accented speech corpus). All VPC baselines use identical data, ensuring fair comparison.
>
> Q3: Incomplete baseline comparisons with VPC systems.
> - We have added all VPC'24 baselines (B2-B6) to Table 3 and Section 5.4 (highlighted). These baselines operate offline without latency constraints, making direct comparison problematic. Offline baselines can leverage full-utterance bidirectional context and computationally expensive multi-pass processing without real-time constraints, whereas streaming systems like TVTSyn must operate causally with <80ms latency, limited lookahead context, and single-pass processing.
> - Additionally, some approaches involve different trade-offs—for example, ASR-TTS cascades (B3) achieve strong anonymization but necessarily remove prosody and emotion, which is acceptable for some applications but problematic when preserving the expressiveness of speech is required.
> - Despite operating with streaming/latency constraints, our system (TVTSyn) achieves competitive privacy (EER=14.57%) and utility (WER=5.35%) when compared to the offline baselines.
> - Section 5.4 explicitly discusses this fairness issue and shows that streaming anonymization need not sacrifice utility.

---

> > ### Comment · Reviewer_1kxu · 2025-11-24
> >
> > Thank you for the authors’ response. However, I find that their rebuttal does not sufficiently address the key concerns raised in my review.
> >
> > - Baselines in Table 3 are not convincing.
> > The authors only compare against the VPC 2024 baseline systems, but do not include any participant systems from the challenge. In practice, the VPC participant systems represent the actual state-of-the-art performance, and excluding them limits the credibility of the comparisons. Without these stronger baselines, the improvements claimed in Table 3 are not fully convincing.
> >
> > - Missing UAR metric for emotion preservation.
> > The lack of an Unweighted Average Recall (UAR) evaluation makes the analysis incomplete. UAR is a standard and necessary metric for quantifying emotion preservation, especially in imbalanced emotion datasets. Its absence weakens the evidence supporting the authors’ claims.
> >
> > Given these unresolved issues, I believe my original assessment remains appropriate and I see no basis for adjusting my original score.

---

> > > ### Author Response · Authors · 2025-11-25
> > > **Rebuttal response from authors**
> > >
> > > Baselines in Table 3 are not convincing. The authors only compare against the VPC 2024 baseline systems, but do not include any participant systems from the challenge. In practice, the VPC participant systems represent the actual state-of-the-art performance, and excluding them limits the credibility of the comparisons. Without these stronger baselines, the improvements claimed in Table 3 are not fully convincing.
> > > - We apologize for the oversight.  Based on your earlier comment *(“Incomplete baseline comparisons with VPC baseline systems or other popular speaker anonymization systems,”)* we had only compared against VPC baselines.  As suggested, we have also added the best VPC'24 participant systems (T10-C3, T9, T8-4, T38-M1) to Table 3.
> > > - Please note that TVTSyn is a ***streaming*** model with low latency (<80ms), whereas all the VPC participant systems operate offline with full-utterance context.  Thus, none of these systems can operate in real-world deployments where offline processing is infeasible.  This is the primary strength of our system.
> > >
> > >
> > > Missing UAR metric for emotion preservation, which makes the analysis incomplete. UAR is a standard and necessary metric for quantifying emotion preservation, especially in imbalanced emotion datasets. Its absence weakens the evidence supporting the authors’ claims.
> > > - As suggested, we have also added UAR metrics to Table 3.  Please note that our goal is ***emotion suppression*** for privacy, not emotion preservation. Thus, in our case, lower UAR is desirable (37%), whereas VPC participant systems were optimized for emotion retention (>60%).  This is, in part, why we had not included UAR in the original submission, i.e., to avoid confusion.
> > >
> > >
> > > The rebuttal does not sufficiently address the key concerns raised in my review.  Given these unresolved issues, I believe my original assessment remains appropriate and I see no basis for adjusting my original score.
> > > - We would like to point out that we had already taken action to resolve two of the three original points (novelty and datasets). We appreciate the opportunity to provide a more comprehensive response to the third concern (comparison against VPC participants) and include UAR measures.

---

### Official Review · Reviewer_VGqB · 2025-11-01

**Soundness:** 4
**Presentation:** 3
**Contribution:** 3
**Rating:** 6
**Confidence:** 4

**Summary:**

The author proposed a streamable speech synthesizer, TVTSyn, for voice conversion and anonymization.
The model resolved a mismtach in prior work that linguistic content is time-varying but speaker identity is a static vector. The proposed model aligns the temporal granularity of identity and content relying on TVT, time-varying timbre representation. It also uses a speech content encoder encoder to extract feature that removes residual speaker information and regularize the content space with a factorized vector-quantized bottleneck.
The experiment shows that the model behaves strong compared to multiple baselines and the system is causal end-to-end with <80ms on GPU and <132ms on CPU, which is considered within real-time bounds. A comprehensive study on content and TVT representation and ablation study is also provided.
Overall, the work has good vision and provide moderate novelty on model architecture and representations to resolve the mismatch between static speaker identity embedding and time-varying timbre representation. The strong empirical results show the effectiveness on model performance including speech naturalness, anonymization and system latency. The limitation of the work is that dataset is english only, evaluation samples size(N=20 for MOS) are limited and the theoretical analysis of the TVT representation could be better discussed.

**Strengths:**

1. The problem is well framed and it shows good intuition on the mismatch between dynamic input and static speaker embedding. The solution is intuitive by introducing a timbre representation that contains better temporal information.
2. The overall system is well designed and end-to-end streamable. The proposed content encoder introduces a learnable bottleneck with factorized vector-quantization(VQ) that learns discrete, speaker-independent units while preserving linguistic fidelity. Also the time varying timbre representation is consists of a global timbre memory (GTM) that allowing content embedding to attend over the keys to retrieve weighted component using attention, and a combination of gating, interpolation to balance stability and flexibility.
3. The author provided a comprehensive evaluation of the Voice Conversion and Speaker Anonimyzation, and latency analysis.  Also it shows the system is 79 ms latency on GPU and around 132 ms on CPU, achieving a real-time.
4. Good analysis on content representation with tSNE visualization, showing the effectiveness of VQ bottleneck. Good representation on ablation study showing the removal of TVT and VQ causes degradation.

**Weaknesses:**

1.The discussion for the design of gating/interpolation are mostly based on intuition and empirical results. It would be good to include more theoretical analysis. These innovations on TVT representation and the usage of gating/slerp interpolation are more at a level of improving on top of existing architectures. Yet they are proved effective from experiment results.
2. The MOS tests are based on 20 samples which is limited and may cause bias

**Questions:**

Could you provide more insights on the design of Factorized VQ and gating/interpolation?

For example, in Section 3.2, you mention that slerp interpolation “respects the hyper-spherical geometry of the embedding space, ensuring smooth trajectories and preserving angular distances” which helps maintain “identity geometry”. Could you clarify or provide theoretical evidence for this claim?
Also in section 5.2 (c) it shows the difference of before and after applying slerp. How sensitive is the output timbre to the choice of interpolation method (Slerp vs. linear)?

---

> ### Author Response · Authors · 2025-11-21
> **We appreciate Reviewer VGqB for the positive assessment and constructive suggestions. We address each concern below.**
>
> We appreciate Reviewer VGqB for the positive assessment and constructive suggestions. We address each concern below.
>
> Q1: Theoretical evidence for Slerp preserving "identity geometry". Analysis mostly intuition based.
> - We have strengthened the theoretical justification in Section 3.2, Gating and Interpolation (highlighted). We explicitly describe Slerp's geometric properties: interpolation along geodesics on the unit hypersphere maintaining constant angular velocity, preserving angular distances and manifold structure. We added new references that support Slerp's superiority in preserving identity characteristics in high-dimensional embeddings: Shoemake (1985) for the foundational formulation, SlerpFace (2024) showing better face identity preservation compared to linear interpolation.
>
> Q2: Sensitivity to interpolation method (Slerp vs. linear).
> - We have added an ablation study (Section 5.3, Table 1) that compares slerp against linear interpolation and shows modest but measurable degradation when using the latter. Privacy (Src-SIM) is largely unaffected, consistent with our finding that privacy is determined by the overall architecture. However, synthesis quality (NISQA) degrades notably (-4.1%), and target similarity decreases slightly (-0.01), indicating that Slerp better preserves speaker identity characteristics while enabling smooth timbre variation.
> Overall, these new results validate our design choice: Slerp provides incremental but consistent improvements in maintaining identity geometry, particularly important for high-quality synthesis where perceptual distortions accumulate across frames.
>
> Q3: MOS sample size (N=20) limited.
> - We acknowledge this limitation in Section 5.1.

---

### Official Review · Reviewer_he7Q · 2025-11-07

**Soundness:** 4
**Presentation:** 3
**Contribution:** 3
**Rating:** 8
**Confidence:** 5

**Summary:**

The paper introduces TVTSyn, a low-latency voice conversion and anonymization system that replaces the traditional static speaker embedding (x-vectors) with a content-synchronous, time-varying timbre (TVT) representation. The method uses a Global Timbre Memory (GTM), gated interpolation (Slerp), and a factorized VQ bottleneck to align the temporal granularity of speaker identity with linguistic content that are fed into a streamable speech synthesizer for generating anonymised audio. The system achieves <80 ms latency on GPU, ~132ms on CPU, shows improved naturalness, and provides a better privacy-utility trade-off than prior streaming baselines (SLT24, DarkStream, GenVC). Experiments follow the VoicePrivacy Challenge 2024 protocol, reporting reasonable performance in source-target speaker similarity, EER, WER, and perceptual quality metrics. EER in semi-informed case is significantly lower than DarkStream which is not elaborated further. It is potentially due to the content embeddings leaking speaker information. I doubt the claim that content embeddings are speaker-independent and needs to be qualified properly.

**Strengths:**

- Authors identify a fundamental weakness (static speaker embedding) in the current speaker anonymisation techniques and proposes a well-justified fix via content-synchronous conditioning of the speaker embeddings
- Well-founded experiments on VPC 2024 protocol and ablations confirm benefits across privacy, quality, and latency
- Deployment conditions are kept in mind by demonstrating real-time performance on CPU/GPU under tight latency budgets (<80 ms), relevant for interactive applications
- Clear figures, well-written text, and reproducible evaluation settings
- Concrete future directions are presented that extend the technique significantly

**Weaknesses:**

- The performance of B1 baseline is mentioned during the analysis but not added to Table 2 for clear comparison
- The gating and Slerp mechanisms are intuitively motivated but not analyzed quantitatively (e.g., contribution to expressivity or privacy).
- Listening tests use a small Mechanical Turk sample (N = 20) without statistical significance analysis or demographic breakdown. A larger cohort of listeners must be recruited (>100) and carefully selected to include demographic variations (age, gender, native/non-native, etc.)
- Authors claim that the content embeddings are speaker-independent and show it through t-SNE plots but do not quantify it through metrics. The claim of speaker-independence needs to be properly verified. One option is to classify speakers directly through content embeddings which might reveal how much speaker information is leaking through them as performed in this paper: https://petsymposium.org/popets/2023/popets-2023-0007.php

**Questions:**

1. Could the authors provide a quantitative ablation showing how the gating parameter $\alpha_t$ or the number of timbre facets $K$ affects privacy (EER) and quality (NISQA) ?
2. Is the Global Timbre Memory fixed per speaker or updated during fine-tuning ? Would dynamic adaptation hurt anonymity ?
3. How robust is this technique to noisy or reverberant inputs ? does the causal encoder maintain intelligibility under such conditions ?

---

> ### Author Response · Authors · 2025-11-21
> **We appreciate Reviewer he7Q for the positive assessment and the constructive feedback. We address each concern below and highlight the revisions made to the manuscript (marked in yellow).**
>
> We appreciate Reviewer he7Q for the positive assessment and the constructive feedback. We address each concern below and highlight the revisions made to the manuscript (marked in yellow).
>
> Q1: Quantitative ablation on gating parameter alpha and number of timbre facets K. Gating and Slerp not analyzed quantitatively.
> - We have added comprehensive ablation studies in Table 1 and Section 5.3 (highlighted). Our findings were as follows. Similarity with the source speaker (Src-SIM) remains constant at 0.48 across all ablations, indicating that privacy is determined by overall architecture rather than fine-grained TVT design and that TVT's primary role is preserving synthesis quality while maintaining anonymization. We observe that removing GTM leads to the largest quality drop (3.91 to 3.45), which indicates that content-synchronous timbre is essential for naturalness. Removing learnable prototypes (priors) also causes a notable degradation (3.91 to 3.62), as the model loses universal timbre bases that enable efficient generalization. Replacing Slerp with linear interpolation and replacing gating with fixed alpha=0.5 produce modest but measurable degradations, validating these design choices. The same is observed when reducing GTM capacity from 48 to 24 tokens or 12 tokens, confirming our choice of 48 tokens balances expressivity and efficiency.
>
> Q2: Is GTM fixed or updated during fine-tuning? Would dynamic adaptation hurt anonymity?
> - The GTM is generated deterministically at inference from the global speaker embedding (see Section 3.2, Equation 1). MLP weights and learnable prototypes are frozen after training and shared across all speakers. Dynamic per-frame adaptation occurs through the attention mechanism selecting relevant facets, not through parameter updates. This design maintains consistency and prevents overfitting to source characteristics.
>
> Q3: Robustness to noisy or reverberant inputs.
> - Though our model was trained on clean data, its robustness can be improved through data augmentation (additive noise, reverberation) as demonstrated in previous work (Ranjan et al., 2024; Lakshminarayana et al., 2025). Systematic noise evaluation is beyond our current scope. We acknowledge this in Section 6 (Discussion) and commit to addressing it in follow-up work.
>
> Q4: The performance of B1 baseline is mentioned during the analysis but not added to Table 2 for clear comparison.
> - Thank you for pointing out this issue. We recognize the confusion arose from inconsistent notation in our baseline system descriptions. We have corrected this in the revised manuscript: Section 4 (Baseline Systems) now uses consistent naming throughout (B1 = SLT24, DS = DarkStream, GenVC-s). Section 5.3 maintains consistent notation when discussing results.
>
> Q5: Listening test sample size (N=20) insufficient.
> - We acknowledge this limitation in Section 5.1.
>
> Q6: Speaker-independence of content embeddings not quantified.
> - We agree speaker classification would strengthen validation. While this extensive experiment is beyond the current revision scope, we have acknowledged the limitation in Section 5.2 and commit to including this analysis in future works.

---

### Author Response · Authors · 2025-12-01
**Official comment by authors**

**Summary for Area Chairs**

Both positive reviewers (he7Q: 8/10 Accept; VGqB: 6/10 Marginally above threshold) recognized sound technical contributions with excellent experimental methodology. Key strengths include: (1) well-motivated problem (static-dynamic mismatch in speaker embeddings), (2) practical streaming architecture achieving <80ms GPU latency while outperforming all streaming baselines, and (3) comprehensive VPC'24 protocol evaluation.

In response to reviewer concerns, we:
- Added comprehensive ablation studies (Table 1, Section 5.3) quantifying TVT component contributions: GTM removal causes -11.8% NISQA degradation (essential for naturalness), learnable prototypes removal causes -7.4% degradation, and Slerp vs. linear interpolation comparison shows -4.1% quality difference.
- Strengthened theoretical justification for Slerp with mathematical derivations showing geodesic preservation on unit hypersphere and supporting citations
- Clarified GTM design (deterministically generated, frozen after training) preventing anonymization leakage


**Regarding Reviewer 1kxu**

We wish to provide context regarding the review process. The reviewer's initial comments were concise. The dataset concerns were unexpected, given our use of established, widely cited speech corpora like LibriTTS. The reviewer also requested baseline comparisons from the Voice Privacy Challenge, which we incorporated into the revised manuscript. We further clarified the novelty of our work by emphasizing that our system is a low-latency streaming model, whereas most previous anonymization systems (except methods already compared in our work) operate off-line after processing an entire utterance.

Although we thoroughly addressed these initial points, the reviewer maintained their original score and introduced new requirements in the second round (e.g., specific VPC participants' results and AUR measures). We implemented these additional changes in the subsequent revision and response.

We have invested significant effort in addressing all feedback comprehensively and believe our manuscript's quality warrants a re-evaluation of its score in light of these substantive revisions

---

### Meta-Review · Area_Chair_LuD4 · 2026-01-03

**Summary:**

The reviewer concerns addressed by the authors, which informed my suggestions, include the need for a quantitative ablation study on the gating parameter and the number of timbre facets, the request for a more rigorous theoretical analysis, and a clearer explanation of the novelty and results.

**Reviewer Concerns:**

The authors have addressed several key reviewer concerns. They added a quantitative ablation study on the gating parameter,  and the number of timbre facets. Moreover, they have improved the theoretical analysis to explicitly describe the geometric properties of Slerp, and have also introduced an ablation study comparing Slerp with linear interpolation. They also addressed concerns regarding insufficient novelty, given prior work on dynamic speaker conditioning in speech synthesis, and incorporated the previously missing VPC’24 baseline systems into the experimental results. Some concerns remain, including the fact that the speaker-independence of content embeddings has not been quantified and that the MOS evaluation is based on a limited sample size (( N = 20 )).

**Reviewer Scores:**

1kxu likely would have maintained their score, but it seems the decision would not have been based on the actual merits of the work, rather on a preconception against it.

VGqB might have slightly increased their score.

he7Q would have kept their score.

---

### Decision · Program_Chairs · 2026-01-26

Accept (Poster)